# Giant voltage-induced modification of magnetism in micron-scale ferromagnetic metals by hydrogen charging

Xinglong Ye [1✉], Harish K. Singh[2], Hongbin Zhang [2], Holger Geßwein[3], Mohammed Reda Chellali[1], Ralf Witte[1], Alan Molinari[1,4], Konstantin Skokov [2], Oliver Gutfleisch [2], Horst Hahn [1] & Robert Kruk[1✉]

Owing to electric-field screening, the modification of magnetic properties in ferromagnetic metals by applying small voltages is restricted to a few atomic layers at the surface of metals. Bulk metallic systems usually do not exhibit any magneto-electric effect. Here, we report that the magnetic properties of micron-scale ferromagnetic metals can be modulated substantially through electrochemically-controlled insertion and extraction of hydrogen atoms in metal structure. By applying voltages of only ~ 1 V, we show that the coercivity of micrometer-sized $SmCo_5$, as a bulk model material, can be reversibly adjusted by ~ 1 T, two orders of magnitudes larger than previously reported. Moreover, voltage-assisted magnetization reversal is demonstrated at room temperature. Our study opens up a way to control the magnetic properties in ferromagnetic metals beyond the electric-field screening length, paving its way towards practical use in magneto-electric actuation and voltage-assisted magnetic storage.

[1] Institute of Nanotechnology, Karlsruhe Institute of Technology, 76344 Eggenstein-Leopoldshafen, Germany. [2] Institute of Materials Science, Technische Universität Darmstadt, 64287 Darmstadt, Germany. [3] Institute for Applied Materials, Karlsruhe Institute of Technology, 76344 Eggenstein-Leopoldshafen, Germany. [4] Present address: IBM Research – Zurich, 8803 Rüschlikon, Switzerland. ✉email: xing-long.ye@kit.edu; robert.kruk@kit.edu

Controlling magnetic properties of magnetic materials by applying small voltages have attracted great interests owing to its ultralow power consumption[1–3]. Most studies so far have been conducted at low temperatures using diluted magnetic semiconductors[4] and multiferroics[5]. By contrast, ferromagnetic metals and intermetallic compounds exhibit high Curie temperatures and large magnetization, making the voltage control of their magnetic properties desirable for room-temperature applications. However, unlike semiconductors and multiferroics, the metallic materials have strong electric-field screening, which makes the voltage control of their magnetic properties difficult. The breakthrough was made by Weisheit and colleagues, who showed that the coerciviy of ultrathin films of FePt(Pd) can be tuned by ~0.005 T by applying voltages to change the electron density at the metal/electrolyte interface[6]. The success of this experiment took advantage of the large surface-to-volume ratio of ultrathin film and the ultrahigh electric field in electrochemical double layer ($>10^9 \, V \, m^{-1}$). However, due to strong electric-field screening, the modification of magnetic properties by this charge-doping method is restricted to a few atomic layers[6–9]. In addition, the changes in magnetic properties are too small for practical use[6,7,9].

Recently, magneto-ionics have been employed to control the magnetic properties of ultrathin metal films. Compared with the charge-doping method, magneto-ionics use ions such as $O^{2-}$ and $H^+$, instead of electrons/holes, as the tuning agent[10–14]. For instance, in the Co (0.9 nm)/GdO$_x$ system[11,12], $O^{2-}$ ions in the ionic-conducting GdO$_x$ layer, driven by the electric field, migrate towards/away from the GdO$_x$/Co interface. The change of oxidation state and crystal structure in the ultrathin Co layer permits to modify its magnetic properties. However, due to electric-field screening in the metal layer, tuning of magnetism via ionic migration is generally limited to the interfacial region within a few atomic layers[10–13]. Although tuning of metallic layer with larger thickness (~15 nm) has also been achieved by magneto-ionics, these tuning processes often suffer from the inherent irreversibility, typical of electrochemical conversion-type reactions[15]. Moreover, the modulation of coercivity reaches only a few tens of mT, thus hindering the practical use of voltage-tuning effect[11,12]. Hence, tuning of the magnetic properties in the volume of ferromagnetic metals by small voltages strong enough from a practical point of view and fully reversible at the same time still remains a challenge.

One, yet unexplored, approach to overcome the electric-field screening limitations is through the insertion and extraction of hydrogen atoms in the metal structure. In the 1970s studies show that some metal and their intermetallic compounds can absorb large amounts of hydrogen atoms that act as hydrogen-storage materials[16,17]. In contrast to electrons and ions, hydrogen atoms are electrically neutral, and therefore their diffusion into the metal structure is not restricted by electric-field screening, offering the opportunity to overcome the limitations of the electric-field screening length in ferromagnetic metals. Moreover, the incorporation of hydrogen atoms often involves the distortion of crystal structure and change of electronic structure, which may change magnetic properties[18,19]. In these studies, however, the absorption of hydrogen atoms was carried out in hydrogen gas usually at high temperature and with high hydrogen pressures. In order to realize the tuning of magnetic properties with small voltages, it would be desirable if the absorption and desorption of hydrogen atoms could be controlled by electrochemical potentials, as established in nickel-metal hydride batteries[19]. Thermodynamically, the hydrogen pressure at certain temperatures can be converted into electrochemical potentials through Nernst equation, which in principle makes the electrochemically controlled hydrogen charging/discharging possible. To test the idea,

we selected SmCo$_5$ as a model material based on two criteria. Firstly, its equilibrium hydrogen pressure is 4 atm at room temperature[15] and according to Nernst equation[20], it is calculated that the equivalent electrochemical potential is only ~17 mV more negative than the standard water electrolysis potential, making SmCo$_5$ suitable for voltage-controlled hydrogen charging/discharging. Secondly, SmCo$_5$ is widely used as an important permanent magnet for its large coercivity, large magnetization ($100 \, A \, m^2 \, kg^{-1}$), and high Curie temperature (1020 K)[21] and is considered candidate material in next-generation ultrahigh density magnetic storage (area density ~10 TB per square inches) because of its exceptionally high magnetocrystalline anisotropy (~17.2 MJ m$^{-3}$)[22]. Tuning of its magnetic properties, particularly magnetocrystalline anisotropy and coercivity with small voltages, would create novel magneto-electric functions in the context of applications.

Here, using micrometer-sized SmCo$_5$ powder, we show that it is possible to reversibly charge and discharge the material with hydrogen atoms by applying small voltages. Employing this approach, the coercivity of SmCo$_5$ powder is tuned by ~1 T, more than two orders of magnitudes larger than previously achieved in ultrathin films by charge doping[6,7,9] and magneto-ionics[10–13]. This enables voltage-assisted magnetization reversal in high-anisotropy SmCo$_5$ at room temperature.

## Results

**Electrochemically controlled charging and discharging with hydrogen atoms.** We used commercially available SmCo$_5$ powders with particle sizes ranging from 1 to 10 μm (Fig. 1a). X-ray diffraction (XRD) showed that the material is single phase with a CaCu$_5$-type hexagonal structure (Supplementary Fig. 1). Transmission electron microscopy revealed no grain boundaries in large particles ~ 10 μm, indicating that the individual particles are single crystals (Supplementary Fig. 2). The saturation magnetization of the powder at 2 T, measured with a superconducting quantum interference device (SQUID) magnetometer, is 100.2 A m$^2$ kg$^{-1}$ at room temperature (Supplementary Fig. 3). This value matches the reported saturation magnetization of SmCo$_5$ (ref. [20]), confirming that the particles are single crystalline. In this measurement, the loose particles were allowed to rotate and align themselves along the magnetic field. For all other magnetic measurements, the SmCo$_5$ particles were fixed by using PVDF binder.

To control the absorption and desorption of hydrogen atoms in SmCo$_5$ with external voltages, we used an electrochemical cell containing an aqueous electrolyte of 1 M KOH (Fig. 1b). The as-prepared SmCo$_5$ electrode and Pt wires were the working and counter electrodes, respectively; the voltage of the working electrode was referenced to Hg/HgO. The process of voltage-controlled charging and discharging with hydrogen atoms can be described as follows. During the absorption (charging), the water molecules at the SmCo$_5$/electrolyte interface were reduced into hydroxide and hydrogen atoms (reaction ① in Fig. 1b). Hydrogen atoms were first adsorbed onto the surface of the SmCo$_5$ particles ($H_{ads}$) and then, driven by the concentration gradient, diffused into the material and were absorbed in either tetrahedral or octahedral sites of the crystal structure ($H_{abs}$, reaction ② in Fig. 1b). Conversely, during the desorption (discharging), the hydrogen atoms on the surface ($H_{ads}$) were oxidized and removed, and then, driven by the gradient of concentration, $H_{abs}$ diffused out, resulting in hydrogen desorption. As shown in the cyclic voltammogram (CV) curve (Fig. 1c), the two current peaks in the cathodic scan correspond to the adsorption and absorption of hydrogen atoms, respectively, whereas the current peak in the anodic scan relates to the desorption (or oxidation) of hydrogen

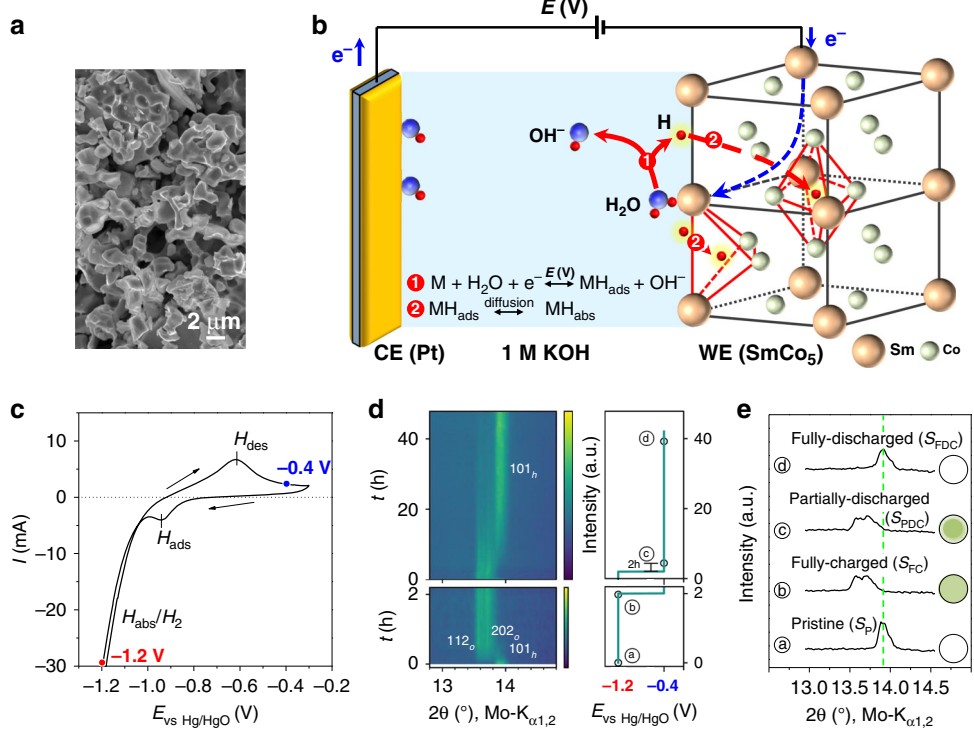

**Fig. 1 Voltage-controlled charging and discharging of SmCo$_5$ with hydrogen atoms. a** An SEM micrograph of the SmCo$_5$ powder, showing particle sizes in the micrometer range. **b** Schematic of the voltage-controlled charging and discharging of hydrogen atoms in the crystal structure of SmCo$_5$. WE working electrode; CE counter electrode. Hydrogen atoms originate from the electrochemical reduction of water molecules at the metal/electrolyte interface (reaction ①). They first adsorb onto the metal surface ($H_{ads}$), and then, driven by the concentration gradient, diffuse into the octahedral or tetrahedral interstitial sites ($H_{abs}$, reaction ②). For clarity, the extraction process is not depicted. **c** A cyclic voltammogram curve of the SmCo$_5$ electrode in 1 M KOH with a scan rate of 2 mV s$^{-1}$. The voltage is with respect to Hg/HgO electrode. **d** Contour plot of in situ XRD patterns of the SmCO$_5$ electrode under the sequential voltages of −1.2 V and −0.4 V. **e** XRD patterns for the samples indicated in **d** with different charging states, i.e. as-prepared (ⓐ, $S_P$), fully charged (ⓑ, $S_{FC}$), partially discharged (ⓒ, $S_{PDC}$) and fully discharged (ⓓ, $S_{FDC}$). The inset schematically illustrates the cross sections of the SmCo$_5$ particle at these charging states.

atoms. The reversible absorption and desorption of hydrogen atoms are similar to that observed in the well-studied LaNi$_5$ used for nickel-metal hydride batteries[23].

The crystal structure of SmCo$_5$ during the charging and discharging was monitored using in situ XRD measurements in transmission mode. The transmission mode allows the detection of the whole volume of the particles rather than only their surfaces. According to Fig. 1c, −1.2 and −0.4 V were applied to induce hydrogen absorption and desorption, respectively. When −1.2 V was applied, the 101 diffraction peak at 13.9° quickly diminished, and two split peaks appeared at 13.7°. The split peaks grew rapidly and remained stable after one hour, indicating that the whole sample was fully charged with hydrogen atoms (Fig. 1d). Mass spectrum confirmed that after charging in 1 M KOH in D$_2$O, the intensity of deuterium peak increased significantly and became comparable to that of hydrogen peaks, confirming the absorption of hydrogen atoms in SmCo$_5$ (Supplementary Fig. 4). The amount of the absorbed hydrogen was determined to be around 2.6 atoms per SmCo$_5$ unit cell by thermogravimetric analysis (Supplementary Fig. 5). Rietveld analysis showed that upon hydrogen insertion the hexagonal CaCu$_5$ structure expanded anisotropically in the basal plane with the c-axis nearly unaffected, revealing the distortion of the original hexagonal structure into an orthorhombic body-centered structure due to hydrogen insertion (Fig. 1b and Supplementary Fig. 6). When the voltage was changed to −0.4 V, the split peaks slowly diminished and the 101 peak started again to develop after ~ 8 h. After a prolonged time of discharging, the 13.9° peak was

completely recovered, indicating that the discharging process was complete.

From the evolution of the XRD patterns, the subsequent stages of the charging and discharging processes can be inferred. When the fully charged sample (at −1.2 V for 1 h, sample ⓑ in Fig. 1d, e) was discharged at −0.4 V for 2 h, the diffraction patterns remained unchanged (sample ⓒ in Fig. 1d, e). This suggests that during the initial stage of the discharging process only the near-surface region of SmCo$_5$ particle was depleted from hydrogen, forming a core-shell structure with the core containing hydrogen atoms (see schematics in Fig. 1e). Hereafter, the different states of the as-prepared SmCo$_5$ sample ($S_P$) after charging at −1.2 V for 1 h and further discharging at −0.4 V for 2 h and 40 h are referred to as fully charged ($S_{FC}$), partially discharged ($S_{PDC}$), and fully discharged ($S_{FDC}$) samples, respectively (Fig. 1e).

**Voltage modulation of coercivity at room temperature.** We explored the response of the magnetic properties of SmCo$_5$ to the applied voltages using in situ SQUID measurements. The coercivity of the $S_P$ sample was ~0.5 T (Fig. 2a). After the sample was fully charged ($S_{FC}$_1st), its magnetization decreased by ~10%. More strikingly, the coercivity decreased by one order of magnitude to ~0.04 T. The observed reductions in the magnetization and coercivity are qualitatively consistent with previous results for RCo$_5$H$_x$ (R = rare-earth metal) synthesized in gaseous hydrogen[24,25]. When the sample was fully discharged ($S_{FDC}$), both the magnetization and the coercivity of the sample were fully

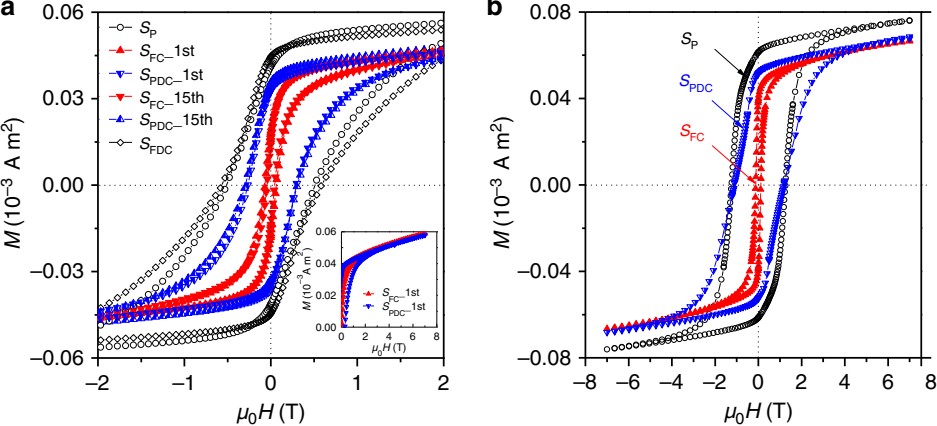

**Fig. 2 In situ voltage control of the coercivity in SmCo$_5$ at room temperature. a** Enlarged hysteresis loops of the as-prepared SmCo$_5$ sample ($S_P$) and those under the applied voltages of $-1.2$ V ($S_{FC}$) and $-0.4$ V ($S_{PDC}$). The $S_P$ sample was repeatedly fully charged (at $-1.2$ V for 1 h, $S_{FC}$) and partially discharged (at $-0.4$ V for 2 h, $S_{PDC}$) for 15 times. Only the hysteresis loops after the 1st and the 15th charging and discharging cycles were shown, revealing excellent reversibility of the coercivity manipulation. The fully discharged sample ($S_{FDC}$) was obtained by discharging the $S_{FC}$ sample at $-0.4$ V for ~40 hours, showing the complete recovery of the coercivity to that of $S_P$ sample. Inset in **a** shows the first quadrant of hysteresis loop of $S_{FC}$_1st and $S_{PDC}$_1st, showing no change of magnetization. **b** Hysteresis loops of the as-prepared SmCo$_5$ sample ($S_P$) and those under the applied voltages of $-1.2$ V ($S_{FC}$) and $-0.4$ V ($S_{PDC}$). In **a** the samples had particle sizes ranging from 1 to 10 μm; those in **b** were ball milled and had smaller particle sizes of ~1 μm.

recovered to their initial values of the pristine ($S_p$) sample (Fig. 2a and Supplementary Fig. 7).

Interestingly, we found that the short-time partial discharging can already significantly increase the coercivity. When $S_{FC}$ was partially discharged at $-0.4$ V for 2 h ($S_{FC}$_1st→ $S_{PDC}$_1st in Fig. 2a), its magnetization remained nearly unchanged (inset in Fig. 2a). This is consistent with in situ XRD measurements detecting no change of the crystal structure (Fig. 1d, e). By contrast, the coercivity increased by a factor of seven from ~0.04 T to ~0.3 T. The reversibility of the coercivity was examined by alternately holding the sample at $-1.2$ V (1 h) and at $-0.4$ V (2 h) to fully charge and partially discharge the sample for 15 times. Magnetic measurements show that after the voltage-switching procedure, the hysteresis loops of both $S_{FC}$_15th and $S_{PDC}$_15th samples overlapped with those before, revealing an excellent reversibility of the voltage-modulation of coercivity in SmCo$_5$ (Fig. 2a). Moreover, the voltage dependence of the coercivity in SmCo$_5$ was studied by treating the individual $S_P$ samples at various voltages from $-0.9$ V to $-1.2$ V and then to $-0.4$ V (Supplementary Fig. 8). As expected, the coercivity changed exactly at the voltages where the hydrogen absorption and desorption occur.

The modulation of the coercivity became larger in magnitude and faster in speed when using the SmCo$_5$ powder with smaller particle sizes of ~1 μm, which displayed a high coercivity of ~1.2 T. After the full charging ($S_{FC}$), the coercivity decreased by one order of magnitude to ~0.1 T. Astonishingly, when the $S_{FC}$ was partially discharged, the coercivity (~1.1 T) was almost fully restored to that of the $S_P$ sample (Fig. 2b). The modulation of coercivity between $S_{FC}$ and $S_{PDC}$ states can be repeated many times. The voltage-driven tuning of the coercivity thus reached an unprecedented value of ~1 T, more than two orders of magnitude larger than those achieved in ferromagnetic metals through charge doping[6,7,9] and magneto-ionics[10–13]. Furthermore, the substantial shortening of the time for the complete recovery of coercivity compared with larger particles suggests that considerable further improvement in the speed can be achieved by reducing the particle sizes or using thin films.

**Voltage-assisted magnetization reversal at room temperature.** With the substantial modulation of the coercivity, the magnetization

reversal can now be assisted by applying low voltages (Fig. 3a). In the measurement, we used the samples with particles sizes of 1–10 μm as those in Fig. 2a. First, we magnetized the $S_{PDC}$ sample with a large magnetic field of $-7$ T (point ① in the inset). Then, the magnetic field was reversed to 0.1 T (point ②). As the 0.1 T field was smaller than the coercive field of the $S_{PDC}$ sample (~0.3 T), the magnetization remained negative and nearly constant until the voltage was changed to $-1.2$ V (point ③). In response to the voltage change, the magnitude of magnetization decreased abruptly and changed from negative to positive in only ~3 min, showing the voltage-assisted magnetization reversal. After ~1.7 h, the magnetization became constant (④). Furthermore, the magnetization reversal can be stopped and reactivated on-demand by switching the applied voltages between $-1.2$ V and $-0.4$ V (Fig. 3b). The "stop and reactivation" process responded to the voltage switching within a few seconds and can be repeated many times without changing the magnetic field. The above results show that the giant modulation of coercivity and the assisted magnetization reversal can be achieved in micrometer-sized SmCo$_5$ by electrochemically controlled hydrogen charging and discharging. Below the possible mechanisms are discussed, starting with the impact of the magnetocrystalline anisotropy constant ($K_1$).

## Discussion

Figure 4a shows the easy and hard axis magnetization curves of the $S_P$, $S_{FC}$, and $S_{PDC}$ samples after aligning and fixing the SmCo$_5$ particles along their easy axis. The value of $K_1$ was obtained by calculating the area enclosed between these two curves (Supplementary Fig. 9). After the full charging, the saturation magnetization ($M_s$) of the $S_P$ sample decreased by ~20%, while $K_1$ decreased by ~40%. The decrease of $K_1$ is qualitatively consistent with the results reported for LaCo$_5$ and CeCo$_5$ after gaseous hydrogenation[26]. Density functional theory (DFT) calculations confirmed that $K_1$ decreased by ~30% when transforming from SmCo$_5$ to SmCo$_5$H$_3$. Moreover, DFT calculations indicated that the decrease in $K_1$ originates from the change of the electrostatic potential around the Sm$^{3+}$ ions with the 4 f charge density essentially unchanged[27] (Supplementary Fig. 10). After hydrogen insertion, the electrostatic potential increased more along the c-axis than along the b-axis (Fig. 4b). This partially canceled the

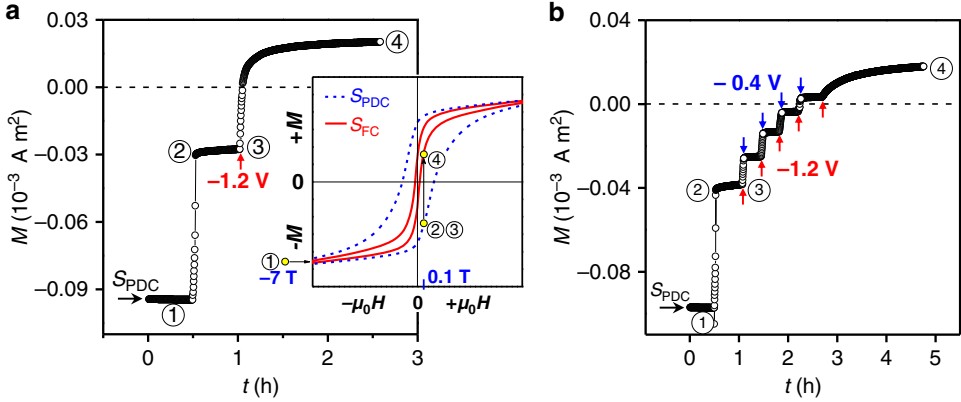

**Fig. 3 Voltage-assisted magnetization reversal in SmCo₅ at room temperature. a** Time evolution of the magnetization in the $S_{PDC}$ sample with the voltages switched from −0.4 V to −1.2 V, showing the voltage-assisted magnetization reversal. Points ①, ②, ③, and ④ indicate different magnetization states as shown in the inset. **b** Time evolution of the magnetization in the $S_{PDC}$ sample with the voltages switched repeatedly between −0.4 V and −1.2 V, showing the voltage-controlled quick and reversible "stop and reactivation" of the magnetization reversal. The red (blue) arrows denote the time points where −1.2 V (−0.4 V) was applied. Here, the same samples as in Fig. 2a were used, with particle sizes of 1–10 μm.

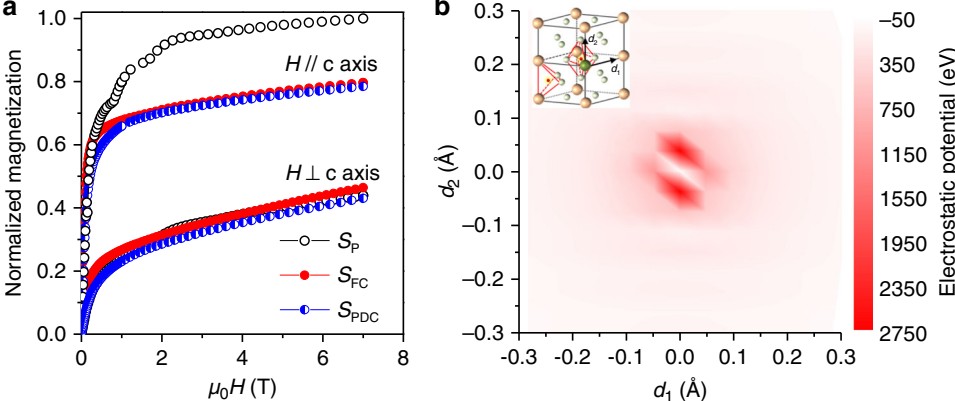

**Fig. 4 Dependence of the magnetocrystalline anisotropy of SmCo₅ on the charging states. a** Easy and hard axis magnetization curves of $S_P$, $S_{FC}$, and $S_{PDC}$ samples. For comparison, the magnetization was normalized by the magnetization value of the $S_P$ sample at 7 T along the easy axis. **b** Contour plot of the difference of the electrostatic potential around $Sm^{3+}$ between SmCo₅ and SmCo₅H₃ within (100) plane. Inset denotes the directions of $d_1$ (along [010]) and $d_2$ (along [001]) in hexagonal structure.

original anisotropy of the electrostatic potential in the $S_P$ sample, resulting in a decrease of both the crystal field and the $K_1$ of the $S_{FC}$ sample.

However, when the $S_{FC}$ sample was partially discharged ($S_{PDC}$), the values of $K_1$ and $M_S$ remained essentially unaffected, since the respective easy and hard axis magnetization curves of $S_{FC}$ and $S_{PDC}$ samples were nearly superimposed on each other (Fig. 4a). Generally, the coercivity of ferromagnetic materials is determined by the intrinsic magnetic parameters ($K_1$ and $M_s$) and the microstructural features, such as the orientation of grains, their sizes, and shapes[28]. During the migration of hydrogen atoms in interstitial sites, these microstructural features of the single-crystalline SmCo₅ particles are expected to remain unaltered, and the change of coercivity should be attributed to the change of $K_1$ and $M_s$. However, the experimental results revealed the nearly unchanged $K_1$ and $M_s$ from the $S_{FC}$ to $S_{PDC}$ states (Fig. 4a). Since $K_1$ and $M_s$ were obtained by measuring the entire volume of the particle, this suggests that the coercivity is not controlled by the entire volume of the particle but rather by its near-surface region. As described earlier, when the $S_{FC}$ sample was partially discharged, only the near-surface regions were actually affected (Figs. 1d, e and 2a). The ability to modulate the coercivity of the whole volume of the particle by only charging and discharging the

near-surface region thus enables the kinetically fast control of coercivity, as verified by the fast reversible stop and reactivation of magnetization reversal in Fig. 3b. The responsible magnetization reversal mechanism regarding the near-surface region in determining the coercivity is next considered.

Because of the high magnetocrystalline anisotropy, the magnetization reversal in single-crystalline SmCo₅ particles is controlled by the nucleation and growth of the reversed magnetic domains from surface defects where local gradient of $K_1$ is capable of significantly lowering the nucleation field[29,30]. The nucleation field can be described by[31]

$$H_n = \frac{1}{2M_s\Delta}\left(\gamma_{SmCo_5} - \gamma_{defect}\right) - DM_s \tag{1}$$

in which $\Delta$ is the width of the transition region where the domain-wall energy changes from $\gamma_{defect}$ at the defects to $\gamma_{SmCo_5}$ in the crystal, and $D$ is the local demagnetizing factor. Since the demagnetizing field (D$M_s$) decreased slightly due to the reduced saturation magnetization ($M_s$) after hydrogen insertion, the huge decrease in coercivity must be attributed to the decrease in the domain-wall energy gradient, $\left(\gamma_{SmCo_5} - \gamma_{defect}\right)/\Delta$. As shown above, $K_1$ of the entire volume of the particle decreased by ~40%

from the $S_P$ to $S_{FC}$ states (Fig. 4). It is reasonable to assume that the $K_1$ of the near-surface region would decrease by the same amount when hydrogen insertion just started. By using $\gamma_{SmCo_5} = 4\sqrt{AK_1}$, in which $K_1$ decreased by ~40% and the exchange constant ($A$) decreased by ~45%[26], the domain-wall energy $\gamma_{SmCo_5}$ in the near-surface region was calculated to decrease by ~43% after hydrogen insertion. If ignoring the negligible value of $\gamma_{defect}$, the value of $\left(\gamma_{SmCo_5} - \gamma_{defect}\right)$ would decrease by ~ 43%. Clearly, the magnetic softening due to the reduction of domain-wall energy is not enough to account for the observed 90% reduction of coercivity from $S_P$ to $S_{FC}$ sample. Additional reduction in the coercivity may originate from the widening of the transition region, Δ. In SmCo$_5$ particles, its size is comparable to local defects, and, usually, it cannot be larger than 10–30 nm[31]. But after hydrogen insertion the inhomogeneous redistribution of hydrogen atoms around the surface defects can be more pronounced, and, consequently, the transition region can be significantly wider[32]. Moreover, when SmCo$_5$ particles were charged with hydrogen atoms, the crystal structure changed from the hexagonal to the orthorhombic one and the unit cell volume expanded by ~5% (Fig. 1d). This may increase the mismatch between the surface defects and the surrounding region, broadening the transition region and reducing the coercivity. Exchange spring effect may also be involved at the soft core/hard shell interfaces. Yet, considering the gradual change of hydrogen concentration from the shell to the core as well as the rather small exchange correlation length of SmCo$_5$ (2–4 nm), the exchange spring effect may be insignificant[33].

In summary, using micrometer-sized SmCo$_5$ as a bulk model material, we show that through electrochemically controlled insertion and extraction of hydrogen atoms in the metal structure, the bulk magnetic properties of ferromagnetic metals can be modulated with giant magnitudes. Our study offers an approach to overcome the limitations of the electric-field screening, opening the door to hugely and reversibly modify the bulk magnetic properties in ferromagnetic metals. This approach should be applicable to many rare earth-transition metal hard magnets, such as NdFeB[34] and Sm$_2$Co$_{17}$ (ref. [35]), as hydrogen diffusion in these materials has been normally observed. In application context, the ability to hugely tune their magnetic properties by applying small voltages, which has not been accessible before, will endow the ferromagnetic metals functions such as in magneto-electric actuation[36], information storage, and processing[37,38]. For instance, the ability to reduce their coercivity temporarily will greatly reduce the energy to demagnetize and reverse the magnetization. Another probable application of our results lies in voltage-assisted magnetic storage. With its exceptionally high magnetocrystalline anisotropy (17.2 MJ m$^{-3}$)[21], SmCo$_5$ can keep the magnetization stable against thermal agitation even when the bit size is 2–3 nm, pushing the area density up to 10 Tb/inch$^2$. However, its use is hindered by the high coercivity, which makes the writing of magnetic bits a problem[38]. The demonstrated ~1 T reduction of coercivity and the voltage-assisted magnetization reversal provides a promising approach to solve this problem, i.e. low-coercivity state for writing and high-coercivity state for long-term storage. It is anticipated that the hydrogen charging/discharging time can be significantly reduced when the material size is reduced to nanometer scale[39–41] and the switching speed can be increased. The production of hydrogen atoms by electrochemical reduction of water molecules is considered much faster than the diffusion of hydrogen atoms in the material. We thus estimated the switching speed at the nanoscale by calculating the diffusion time according to the diffusion equation $l = \sqrt{Dt}$, in which $l$ is the diffusion length, $D$ the diffusion coefficient and $t$ the diffusion time. In Fig. 1d, the charging/discharging time of SmCo$_5$ particles

with sizes of 1–10 μm is ~10 min/40 h. Therefore, the diffusion time for a thin film at the nanometer scale can be expected to be reduced by several orders of magnitudes to ms and sub-ms range, which is comparable to the fastest switching speed (~1 ms for 1-nm thick cobalt layer)[38] achieved by magneto-ionics at similar length scales. In addition, based on this equation, the calculated diffusion coefficient falls in the range of $10^{-8}$–$10^{-13}$ cm$^2$ s$^{-1}$ at room temperature, still much smaller than that obtained in gaseous hydrogen ($10^{-8}$–$10^{-10}$ cm$^2$ s$^{-1}$)[42,43], indicating that significant improvements in switching speed may be achieved by optimizing the electrochemical-cell (device) geometry[44,45]. Furthermore, the diffusion of hydrogen atoms can be speeded up significantly by including high diffusion paths such as grain boundaries. This could be especially exciting for use in neuro-morhpic computing, where the slow switching rate of ~100 Hz and the large tuning magnitude are needed[46].

## Materials and methods

**Materials and microstructure characterization.** The SmCo$_5$ powders were purchased from Alfa Aesar (Stock No. 42732.18). The composition of the powder was analyzed by inductively-coupled plasma mass spectroscopy (Supplementary Table 1) and its microstructure was characterized using field-emission scanning electron microscope (Zeiss Ultra 600), powder X-ray diffraction with a Mo K$_\alpha$ source (Philips X'Pert Analysis) and transmission electron microscope (FEI Titan 80-300). The preparation of TEM samples followed the ordinary procedure of cutting, lifting and milling using FIB/SEM system (FEI Strata 400 and Zeiss Auriga 60). For the magnetic measurements in Fig. 2b, the as-received powder was vibration-milled for 1 h (Retsch MM400) and sieved to reduce the particle size. For all other measurements, the as-received powders were used.

**Preparation of the SmCo5 electrode and the electrochemical set-up.** To prepare the SmCo$_5$ electrode, the SmCo$_5$ particles were mixed with PVDF solution to form slurry, which was then coated onto thin copper foils (thickness ~15 μm). The slurry/Cu foil composite was dried at 80 °C for 4 h, and afterward compressed under a pressure of ~100 MPa to further fix the particles and to increase the electrical conductivity between SmCo$_5$ particles and the Cu foil. We prepared the PVDF solution by dissolving PVDF powder in NMP solution at a mass ratio of 5:95 with overnight stirring.

The charging and discharging of the SmCo$_5$ electrodes were carried out under potentiostatic control in a three-electrode electrochemical system (Autolab PGSTAT 302N). The working, counter, and reference electrodes were the SmCo$_5$ electrode, Pt wires and a pseudo Ag/AgCl electrode, respectively. The potential of the peuso Ag/AgCl electrode is 0.300 ± 0.002 V more positive than the standard Hg/HgO (1 M KOH) electrode, and for comparison, all the voltages in the paper were converted to the Hg/HgO scale. The electrolyte was an aqueous electrolyte of 1 M KOH prepared from ultrapure water with a resistivity of ~18.2 MΩ.

**In situ XRD measurement.** The crystal structure of the SmCo$_5$ electrode under the application of −1.2 V and −0.4 V was monitored by in situ XRD with a parallel beam laboratory rotating anode diffractometer (Mo K$_\alpha$ radiation) in transmission geometry. The transmission geometry allowed the detection of the entire volume of the SmCo$_5$ particles rather than only their surfaces. For in situ measurement, the SmCo$_5$ electrode, as the working electrode, was attached to a glass plate (thickness ~0.1 mm) and then immersed in the 1 M KOH electrolyte contained in plastic bags. The counter and reference electrodes were the Pt wire and the pseudo Ag/AgCl electrode, respectively. Diffraction patterns were collected every 371 seconds with a Pilatus 300K-W area detector. The function NIST SRM660b LaB$_6$ powder was used for the detector calibration and determination of the instrumental resolution. The 2D diffraction images were integrated using the pyFAI software and analyzed with the Rietveld method (TOPAS V6). The isostructural orthorhombic β$^{II}$ structure of PrCo$_5$H$_3$ (Im2m space group symmetry) was used as a structure model for the SmCo$_5$ after hydrogen insertion.

**In situ SQUID measurement.** In situ magnetic measurement was carried out with a custom-built miniaturized Teflon electrochemical cell in a superconducting quantum interference device (SQUID, MPMS3) at room temperature. In the electrochemical cell, the SmCo$_5$ electrode, Pt foil and peuso Ag/AgCl electrode were the working, counter and reference electrodes, respectively. The electrolyte was 1 M KOH. The SmCo$_5$ electrode and the Pt foil were attached to the flat surface of a plastic rod, while the reference electrode was threading through a capillary. The magnetic measurements were performed at the sealed mode of SQUID. All magnetic measurements were performed with the applied magnetic field parallel to the surface of the Cu foil.

For the determination of magnetocrystalline anisotropy constant ($K_1$), the SmCo$_5$ particles were first aligned in a homogeneous magnetic field before the

drying of the slurry/Cu foil composite. Other steps in the preparation of the SmCo$_5$ electrode were the same as those described earlier. Before the magnetic measurements in SQUID, the SmCo$_5$ electrode was demagnetized with the vibrating fields from a value of 7–0 T. The sample was first measured along the easy axis. Then, the sample was removed from the plastic rod and remounted in a perpendicular direction and measured again. According to ref. [37], $K_1$ was calculated by integrating the area enclosed between the hard and easy axis magnetization curves. Since the applied magnetic field was way smaller than the anisotropy field of SmCo$_5$ (~40 T), these two curves were extrapolated until they met and then the enclosed area was calculated.

**TG and APT measurement**. Thermogravimetric (TG) measurement was conducted in a Sensys Evo TG-DSC apparatus (Setaram). The as-prepared SmCo$_5$ electrode was fully charged at −1.2 V for 1 h. To remove the residual water, the fully charged sample was rinsed into dehydrated acetone several times, and transferred into the TG chamber after the drying for the TG measurement. To analyze the evolved gas, mass spectrometry was carried out simultaneously with an OmniStar (Pfeiffer, Germany). During the measurements, the temperature was ramped at a rate of 5 °C min$^{-1}$ to 80 °C and then held for another 2 h.

For the atom probe tomography (APT) measurement, the SmCo$_5$ electrode was charged at −1.2 V for 1 h in 1 M KOH in D$_2$O using the three-electrode system as described earlier. After the full charging, the sample was transferred to FIB/SEM system for the cutting, milling and lifting at room temperature (Zeiss Auriga 60). To refine the APT tip, annular milling was used to create the needle-shaped morphology with a diameter less than ~100 nm. APT measurements were conducted on a CAMECA-LEAP 4000×HR instrument in laser pulse mode (wave length 355 nm, pulse frequency 100 kHZ, pulse energy 60 pJ, evaporation rate 0.50%) at a specimen temperature of 20 K. APT reconstruction and analysis were carried out using the CAMECA IVAS version 3.6.1 software.

**DFT calculation**. DFT calculations were performed using all electron full potential local orbital (FPLO) code version 18.00-52 (ref. [47]). The exchange-correlation energy functional was approximated using the generalized gradient approximation within Perdew−Burke−Ernzerhof parameterization[48]. A linear tetrahedron method was used for the $k$-space integration with Blöchl corrections. $k$-point meshes of $8 \times 8 \times 10$ and $10 \times 10 \times 10$ were used for the SmCo$_5$ and SmCo$_5$H$_3$ samples, respectively. The 4f electrons of Sm$^{3+}$ have been treated within atomic limit approach (LSDA + U). The magnetocrystalline energies (MAE) were calculated using the full relativistic mode. After checking the dependence of MAE on U, the desired MAE was obtained with $U = 8$ and 5 eV for SmCo$_5$ and SmCo$_5$H$_3$, respectively. The lattice constant and atomic positions were relaxed for SmCo$_5$ and SmCo$_5$H$_3$. The optimized lattice constants and volume agreed well with the experimental values within a discrepancy of ~1%.

## Data availability
All data needed to evaluate the conclusions of the study are present in the paper or the supplementary materials.

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

## Acknowledgements

We thank Guiying Tian for assisting the preparation of the SmCo₅ electrode, Zhenyou Li, Zhirong Zhao-Karger for thermogravimetric measurements, Wu Wang for TEM characterization, Thomas Schrefl for stimulating discussion, and Ke Lu, Virgil Provenzano for critical reading of the manuscript. The Lichtenberg high performance computer of the TU Darmstadt and KNMF are acknowledged. This study is supported by Deutsche Forschungsgemeinschaft under contract number HA 1344/34-1 (H.H., R.K, R.W., A.M.,); Alexander von Humboldt Foundation & Helmholtz-OCPC Association (X.Y.), Deutsche Forschungsgemeinschaft through CRC/TRR 270, Project ID 405553726 (K.S., O.G.). We acknowledge support by the KIT-Publication Fund of the Karlsruhe Institute of Technology.

## Author contributions

X.Y, R.K. and H.H. conceived the project. X.Y. designed and performed the experiments. H.K.S. and H.Z. conducted DFT calculations with results analysis. H.G. and X.Y. performed in situ XRD measurement. M.R.C. performed APT measurements and analysis. K.S. and X.L. performed magnetic alignment of particles. R.W. assisted the in situ magnetic measurements. X.L. and R.K. interpreted the results with input from K.S. and O.G. X.L. wrote the paper and all authors revised the paper.

## Funding

## Competing interests

The authors declare no competing interests.
