## [Peer Review File · Nature Communications]

Reviewers' Comments:

Reviewer #1:

Remarks to the Author:

The manuscript submitted by X. Ye et al. entitled "Giant Voltage-Induced Modification of Magnetism in Bulk Ferromagnetic Metals by Hydrogen Charging" is well-written and convincing, with high-quality data, thorough analysis, and an extensive amount of supporting information. I consequently find myself in the unusual position of having no major questions regarding the principal conclusions of the manuscript.

That being said, I am having a difficult time convincing myself that this work represents the sort of advance in the field that would warrant publication in Nature Communications. By the authors' own admission in the thorough background and introduction, the effects of hydrogen incorporation into RCo5 systems is well known, as are the electrochemical and diffusive effects used to drive the hydrogen redistribution. The authors make two main claims for impact:

1. That this technique will "open up a way to control the macroscopic magnetic properties in ferromagnetic metals". I think this is extremely unlikely - even switching in a 10 micron particle took hours and is fundamentally limited by the rate of concentration-gradient induced hydrogen diffusion in the metal.
2. That this technique may be readily adapted at the nanoscale in thin films and nanoparticles to enable more rapid switching for field assisted switching. Here, I think the authors are likely correct but it is unclear what the advantages are over already extant approaches using magneto-ionics where switching should inherently be faster due to the charged ions used to accomplish the same goal.

I do not wish to be overly critical here - I found this to be an interesting read and very high-quality work. It should absolutely be published in a form very similar to the present draft. I'm just not convinced it will draw broad enough interest for publication in Nature Communications as opposed to a more specialized journal.

In the event that the authors choose to revise the work, my main suggestions for the actual content of the work is that it was sometimes unclear which experiments and figures referred to which particle sizes. Thus, I was sometimes unsure how to evaluate size effects and switching speeds. I'd make it explicit as much as possible. Secondly, the mass spec work shown in Figure S4 was quite important and I think it could be referred to in more detail in the main text rather than relegated to the supporting information.

Reviewer #2:

Remarks to the Author:

This is an interesting manuscript that reports on the effects of voltage on the magnetic properties of electrolyte-gated SmCo5 particles (mediated through voltage-driven hydrogen absorption).

This work opens new avenues for magnetoelectric actuation of ferromagnetic materials. It can have impact in applications dealing with hard magnetic materials, spintronics, iontronics and any magnetically actuated device, where the reported drastic reduction of coercivity would significantly reduce the energy power consumption. It also explores a new mechanism to induce magnetoelectric effects in thick ferromagnetic films/particles.

Thus, the paper is of high interest for the scientific community (both from fundamental and applied viewpoints) and I recommend its publication. However, there are a few minor issues/comments the authors should address to further improve the quality of the manuscript:

- i) The authors claim that the reported procedure is suitable to tune the magnetic properties of "bulk" ferromagnets. I find this statement a bit far-fetched since they investigate the changes in

the magnetic properties of micrometer-sized particles. I understand the point raised by the authors since, typically, electric field effects in metallic ferromagnets are limited to few nm thick films. Yet, for the general audience (not necessarily familiar with magnetoelectricity), to refer to micrometer-sized particles as "bulk" can be misleading. I would rather use, perhaps, "thick particles" or, at most, "bulky particles".

ii) Can the authors estimate how deep can hydrogen atoms be introduced (absorbed) inside the SmCo₅ particles? Could this be supported by some calculations, inferred from structural characterization, or guessed from magnetic measurements? Would the effect still work, for example, in particles of 100 microns or 1 mm size?

iii) Do the authors envisage that such effects might also be observed in other hard magnetic materials? Why did they choose specifically SmCo₅, for example?

iv) Such hard magnetic materials contain rare-earths (Sm in this case), which are rather prone to oxidation. Did the samples become partially oxidized during the magnetoelectric experiments? Did the authors use some procedure to prevent from oxidation? Why there is apparently no oxidation? The authors could explain this in terms of the voltage values applied.

v) Can the authors comment a bit more on the dynamics of the system (for example, for the cycling experiments)? Being H atoms very light, I might be expecting faster rates (as in the works by Prof. Beach -although those were for ultra-thin films). The magneto-ionic rates here seem comparable to oxygen or lithium ion intercalation. Can the authors compare with other typical magneto-ionic rates on thick films?

vi) Can the initial pristine coercivity be recovered? Why not? It seems there is some irreversible structural change in the particles the first time voltage is applied. Can the authors further comment on that?

vii) One of the intriguing aspects of the work (also pointed out by the authors) is why the coercivity changes occur much faster than those of the magnetization when the sample is subject to applied voltage. Besides the explanations provided by the authors (which seem plausible to me, in spite of the complexity of the effects), could it be that there is some exchange-spring mechanism going on here as well, similar to that of "spring magnets" but inducing softening (rather than hardening)?

REVIEWER COMMENTS (NCOMMS-20-09095-T)

Reviewer #1 (Remarks to the Author):

The manuscript submitted by X. Ye et al. entitled "Giant Voltage-Induced Modification of Magnetism in Bulk Ferromagnetic Metals by Hydrogen Charging" is well-written and convincing, with high-quality data, thorough analysis, and an extensive amount of supporting information. I consequently find myself in the unusual position of having no major questions regarding the principal conclusions of the manuscript.

That being said, I am having a difficult time convincing myself that this work represents the sort of advance in the field that would warrant publication in Nature Communications. By the authors' own admission in the thorough background and introduction, the effects of hydrogen incorporation into RCo₅ systems is well known, as are the electrochemical and diffusive effects used to drive the hydrogen redistribution. The authors make two main claims for impact:

1. That this technique will "open up a way to control the macroscopic magnetic properties in ferromagnetic metals". I think this is extremely unlikely - even switching in a 10 micron particle took hours and is fundamentally limited by the rate of concentration-gradient induced hydrogen diffusion in the metal.
2. That this technique may be readily adapted at the nanoscale in thin films and nanoparticles to enable more rapid switching for field assisted switching. Here, I think the authors are likely correct but it is unclear what the advantages are over already extant approaches using magneto-ionics where switching should inherently be faster due to the charged ions used to accomplish the same goal.

I do not wish to be overly critical here - I found this to be an interesting read and very high-quality work. It should absolutely be published in a form very similar to the present draft. I'm just not convinced it will draw broad enough interest for publication in Nature Communications as opposed to a more specialized journal.

In the event that the authors choose to revise the work, my main suggestions for the actual content of the work is that it was sometimes unclear which experiments and figures referred to which particle sizes. Thus, I was sometimes unsure how to evaluate size effects and switching speeds. I'd make it explicit as much as possible. Secondly, the mass spec work shown in Figure S4 was quite important and I think it could be referred to in more detail in the main text rather than relegated to the supporting information.

Reviewer #2 (Remarks to the Author):

This is an interesting manuscript that reports on the effects of voltage on the magnetic properties of electrolyte-gated SmCo₅ particles (mediated through voltage-driven hydrogen absorption).

This work opens new avenues for magnetoelectric actuation of ferromagnetic materials. It can have impact in applications dealing with hard magnetic materials, spintronics, iontronics and any magnetically actuated device, where the reported drastic reduction of coercivity would significantly reduce the energy power consumption. It also explores a new mechanism to induce magnetoelectric effects in thick ferromagnetic films/particles.

Thus, the paper is of high interest for the scientific community (both from fundamental and applied viewpoints) and I recommend its publication. However, there are a few minor issues/comments the authors should address to further improve the quality of the manuscript:

i) The authors claim that the reported procedure is suitable to tune the magnetic properties of "bulk" ferromagnets. I find this statement a bit far-fetched since they investigate the changes in the magnetic properties of micrometer-sized particles. I understand the point raised by the authors since, typically, electric field effects in metallic ferromagnets are limited to few nm thick films. Yet, for the general audience (not necessarily familiar with magnetoelectricity), to refer to micrometer-sized particles as "bulk" can be misleading. I would rather use, perhaps, "thick particles" or, at most, "bulky particles".

ii) Can the authors estimate how deep can hydrogen atoms be introduced (absorbed) inside the SmCo₅ particles? Could this be supported by some calculations, inferred from structural characterization, or guessed from magnetic measurements? Would the effect still work, for example, in particles of 100 microns or 1 mm size?

iii) Do the authors envisage that such effects might also be observed in other hard magnetic materials? Why did they choose specifically SmCo₅, for example?

iv) Such hard magnetic materials contain rare-earths (Sm in this case), which are rather prone to oxidation. Did the samples become partially oxidized during the magnetoelectric experiments? Did the authors use some procedure to prevent from oxidation? Why there is apparently no oxidation? The authors could explain this in terms of the voltage values applied.

v) Can the authors comment a bit more on the dynamics of the system (for example, for the cycling experiments)? Being H atoms very light, I might be expecting faster rates (as in the works by Prof. Beach -although those were for ultra-thin films). The magneto-ionic rates here seem comparable to oxygen or lithium ion intercalation. Can the authors compare with other typical magneto-ionic rates on thick films?

vi) Can the initial pristine coercivity be recovered? Why not? It seems there is some irreversible structural change in the particles the first time voltage is applied. Can the authors further comment on that?

vii) One of the intriguing aspects of the work (also pointed out by the authors) is why the coercivity

changes occur much faster than those of the magnetization when the sample is subject to applied voltage. Besides the explanations provided by the authors (which seem plausible to me, in spite of the complexity of the effects), could it be that there is some exchange-spring mechanism going on here as well, similar to that of "spring magnets" but inducing softening (rather than hardening)?

Response to reviewers' comments

Reviewer #1 (Remarks to the Author):

The manuscript submitted by X. Ye et al. entitled "Giant Voltage-Induced Modification of Magnetism in Bulk Ferromagnetic Metals by Hydrogen Charging" is well-written and convincing, with high-quality data, thorough analysis, and an extensive amount of supporting information. I consequently find myself in the unusual position of having no major questions regarding the principal conclusions of the manuscript.

Re: We greatly appreciate the reviewer's highly-favorable comments on the quality of our study. We conceive your comments as an encouragement for pursuing solid and thorough studies. We hope that you may find all your questions and comments addressed thoroughly and adequately.

That being said, I am having a difficult time convincing myself that this work represents the sort of advance in the field that would warrant publication in Nature Communications. By the authors' own admission in the thorough background and introduction, the effects of hydrogen incorporation into RCo₅ systems is well known, as are the electrochemical and diffusive effects used to drive the hydrogen redistribution.

Re: We have to admit that after reading these comments we realized that the introduction may not reflect the actual state-of-the-art, and may even be misleading for the reader as to the motivation behind our research. The research field – magnetic properties versus electrochemically-controlled hydrogen charging/discharging – is not at all developed and there is no coordinated research in this direction. To put our study in a proper perspective one can state that there has been no connection made in the literature between electrochemical hydrogen charging/discharging and the control of magnetic properties: these particular research fields have remained separated.

The initial state of the research fields relevant to our work can succinctly be summarized in three points: *i*) In 1970s, researchers found that some intermetallic compounds of the composition AB₅ (A-rare-earth metal, B-transition metal) can absorb large amounts of hydrogen atoms. The main motivation of those studies is the search for hydrogen-storage materials and they did not pay much attention to how and why hydrogen incorporation changes magnetic properties. *ii*) In those studies, the hydrogen absorption/desorption is carried out in hydrogen gas usually at high temperatures and with high hydrogen pressure. *iii*) In nickel-metal hydride batteries, hydrogen absorption/desorption is controlled electrochemically.

It is very telling that no research on the electrochemically-controlled hydrogen absorption/desorption in hard magnets have been reported, not to mention the tuning of the coercivity and magnetization

reversal in hard magnets with small voltages. In our work, we could not rely on well-established knowledge or even scattered hints in the literature; we had to start from scratch, we had to determine the model material, the electrochemical conditions and experimental procedures, to study how hydrogen charging/discharging changes magnetic properties and elucidate the fundamental mechanism behind.

One important aspect of our original idea was to effectively combine electrochemically-controlled hydrogen charging/discharging with the distinctive magnetic properties in metallic hard magnets. To realize the voltage-control of magnetic properties in metals we needed to merge diverse research fields including hydrogen storage, electrochemistry, and magnetism. In our opinion an exceptionally strong magnetoelectric effect demonstrated in metal ($\Delta H_c \sim 1$ T by 1 V) resulting from the combination of conventionally diverse concepts will attract a much broader readership that one can expect in more specialized journal. One could venture to summarize this work saying that “by electrochemically-controlled hydrogen charging/discharging, we have successfully overcome the electric-field screening in metals and produced giant magnetoelectric effect “.

To better reflect the state-of-the-art and convey the difference of our study from the literature, we have revised the third paragraph in page 2 to:

One, yet unexplored, approach to overcome the electric-field screening limitations is through the insertion and extraction of hydrogen atoms in metal structure. In 1970s studies show that some metal and their intermetallic compounds can absorb large amounts of hydrogen atoms that act as hydrogen-storage materials¹⁵⁻¹⁷. In contrast to electrons and ions, hydrogen atoms are electrically neutral, and therefore their diffusion into the metal structure is not restricted by electric-field screening, offering the opportunity to overcome the limitations of the electric-field screening length in ferromagnetic metals. Moreover, the incorporation of hydrogen atoms often involves the distortion of crystal structure and change of electronic structure, which may change magnetic properties. In these studies, however, the absorption of hydrogen atoms was carried out in hydrogen gas usually at high temperature and with high hydrogen pressures. In order to realize the tuning of magnetic properties with small voltages, it would be desirable if the absorption and desorption of hydrogen atoms could be controlled by electrochemical potentials, as established in nickel-metal hydride batteries¹⁹. Thermodynamically, the hydrogen pressure at certain temperatures can be converted into electrochemical potentials through Nernst equation, which in principle makes the electrochemically-controlled hydrogen charging/discharging possible. To test the idea, we selected SmCo_5 as a model material based on two criteria. Firstly, its equilibrium hydrogen pressure is 4 atm at room temperature¹⁸ and according to Nernst equation, it is calculated that the equivalent electrochemical potential is only ~ 17 mV more negative than the standard water electrolysis potential, making SmCo_5 suitable for voltage-controlled hydrogen charging/discharging. Secondly, SmCo_5 is widely used as an important permanent magnet for its large coercivity, large magnetization (100 A m²/kg) and high Curie temperature (1020 K)¹⁹ and is considered candidate material in next-generation ultrahigh density magnetic storage (area density ~ 10 TB/inch²) because of its exceptionally high magnetocrystalline anisotropy (~ 17.2 MJ/m³)²⁰. Tuning of its magnetic properties,

particularly magnetocrystalline anisotropy and coercivity with small voltages, would create novel magneto-electric functions in the context of applications. Here, using micrometer-sized SmCo_5 powder, we show that it is possible to reversibly charge and discharge the bulk of the material with hydrogen atoms by applying small voltages. Employing this approach, the coercivity of SmCo_5 powder was tuned by ~ 1 T, more than two orders of magnitudes larger than previously achieved in ultrathin films by charge doping^{6,7,9} and magneto-ionics¹⁰⁻¹³. This enabled voltage-assisted magnetization reversal in bulk SmCo_5 at room temperature.

The authors make two main claims for impact:

We understand that the reviewer has doubts about the potential impact of our approach in terms of the switching speed of the system and its advantages (or rather lack of these) over the existing magnetoionics. We address these concerns by answering more specific questions below.

1. That this technique will "open up a way to control the macroscopic magnetic properties in ferromagnetic metals". I think this is extremely unlikely - even switching in a 10 micron particle took hours and is fundamentally limited by the rate of concentration-gradient induced hydrogen diffusion in the metal.

Re: Regarding the tuning speed, we agree with the reviewer that it is controlled by hydrogen diffusion in the material. Yet, we want to point out that one major finding of our study is that it is only necessary to charge/discharge the near-surface region rather than the whole volume of the SmCo_5 magnet to tune its coercivity (bulk property), as described in paragraph 3, page 4 and by the discussion of the mechanism behind coercivity change.

This would greatly reduce the necessary diffusion length of hydrogen atoms in SmCo_5 sample, and, therefore, reduce the switching time. As a consequence, one needs only a few seconds to tune the coercivity in 1~10 μm particles to change the coercivity, as evidenced by the fast "stop and reactivation" of magnetization reversal in Fig. 3. How deep hydrogen has to go to bring about the volume magnetoelectric effect is an open question, which goes beyond the purpose of the present manuscript, but we hope to tackle it in the following study on thin films. In addition, the diffusion of hydrogen atoms, even into much larger volumes, can be speeded up significantly by including high diffusion paths such as grain boundaries or other defects.

We agree with the reviewer that the word "macroscopic" is less accurate and removed it from the manuscript. We intend to stress that our electrochemically-controlled hydrogen charging/discharging approach can overcome the electric-field screening length and allows for tuning magnetic properties in the bulk of ferromagnetic metals rather than only a few monolayers at the surface. In addition, it is generally accepted that the size at which materials display different properties compared with bulk materials is around 100 nm, and, therefore, materials larger than 100 nm are often defined as bulk materials. Moreover, as in these magnets like SmCo_5 , nucleation and growth of reversed domains, typically avalanche-type events, are often dominating magnetization reversal, and, therefore, a switching on the length scales described here might actually be sufficient to tune the coercivity of

entire blocks of magnets (\gg mm). Hence, we think that it may be acceptable to keep the word “bulk” in the manuscript. To clarify, we explained the word “bulk” with following words “beyond the electric-field screening length”. The revised sentence in abstract now reads:

“open up a way to control the bulk magnetic properties of ferromagnetic metals beyond the electric-field screening length”. In addition, in page 6, line 9, the sentence “the ability to modulate the macroscopic coercivity” has been changed to “the ability to tune the coercivity of the whole volume of the particle”.

2. That this technique may be readily adapted at the nanoscale in thin films and nanoparticles to enable more rapid switching for field assisted switching. Here, I think the authors are likely correct but it is unclear what the advantages are over already extant approaches using magneto-ionics where switching should inherently be faster due to the charged ions used to accomplish the same goal.

Re: We appreciate the reviewer’s comment. Regarding the switching speed under optimal conditions (size, surface quality, geometry etc.) we would like to argue with the reviewer’s statement “*already extant approaches using magneto-ionics where switching should inherently be faster due to the charged ions*”.

We compared the switching speed of our approach with that of magneto-ionics at the similar length scale. To the best of our knowledge, the fastest switching speed of magneto-ionics is by far achieved by Prof. Geoffrey Beach’s group, and, therefore, we refer to their work to make comparisons.

In their papers (ref. 11,12), for Co/GdO_x system with Co layer of 0.9 nm, to tune the coercivity by a few tens of mT, the tuning time with O²⁻ is 100~ 200 s at 100 °C (Fig. 2, 3 in ref. 11). By increasing the temperature or engineering the electrode and GdO_x, the tuning time can be reduced to 10 ms at 120 °C (Fig. 3f in ref. 11) or to 700 μs at 100 °C (Fig. 4i in ref. 11). By using the voltage-driven migration of H⁺, for the GdO_x/Co system with Co thickness of 0.9 nm, the so-far achieved fastest tuning time is 100-400 ms at room temperature (Supplementary Fig. 10 in ref. 12). The switching speed of magneto-ionics is mainly limited by the slow diffusion of ions in the conducting oxide. Below we compared the room-temperature tuning speed of magnetoionics with that of our approach.

To estimate the switching speed of our approach, we calculated the diffusion time of hydrogen atoms in SmCo₅ with a comparable thickness of 1 nm. The diffusion equation relates the diffusion length (l) to diffusion time (t) by

$$l = \sqrt{Dt} \quad [1]$$

in which D is the diffusion coefficient. As can be seen from *in situ* XRD contour plot (Fig. 1D), for particles with sizes of 1~10 μm, it takes only about 10 minutes for the complete charging of the whole particle. For estimation, we use the charging time of 10 minutes and the particle size of 5 μm. Then, assuming the same diffusion coefficient, it is calculated that for a thickness of 1 nm, the diffusion time is approximately 2.4×10^{-5} seconds (0.024 ms) according to equation [1]. This is at least three orders of magnitude faster than the fastest rate achieved by magneto-ionics at room

temperature (100-400 ms in ref 12). Even if we use the much longer discharging time of 40 hours to calculate, the diffusion time is 5.8×10^{-3} seconds, still at least one order of magnitude faster than magneto-ionics at room temperature.

To reflect the comparison of the tuning speed of our approach with magneto-ionics, we have added a few sentences to the conclusion of the manuscript (page 7, line 28) and it reads:

It is anticipated that the hydrogen charging/discharging time can be significantly reduced when the material size is reduced to nanometer scale and the switching speed can thus increase. To estimate, we calculated the diffusion time in SmCo_5 with a thickness of 1 nm according to the diffusion

equation $l = \sqrt{Dt}$, in which l is the diffusion length, D diffusion coefficient and t diffusion time. In

Fig. 1D, the charging time of SmCo_5 particles with sizes of 1~10 μm is about 10 minutes, and then the diffusion time for a thin film with a thickness of 1 nm is calculated to be around 0.02 ms at room temperature, at least three orders of magnitudes faster than magneto-ionics. In addition, the diffusion of hydrogen atoms can be speeded up significantly by including high diffusion paths such as grain boundaries.

To emphasize the advantage of our approach against magneto-ionics we would like to first clarify their fundamental differences and then the consequent tuning effects, which are briefly summarized in the figure below.

In magneto-ionics, the ions exist in the ionic conducting oxide, and driven by electric field in the oxide, they move towards/away from the ferromagnetic metal/ionic oxide interface. This changes the oxidation state of ferromagnetic metal layer and thereby its magnetic properties. Due to the electric-field screening in metals, magneto-ionics is limited to only a few monolayers at the metal

surface. For instance, in the papers that Prof. Geoffrey Beach's group firstly invented the magneto-ionics, they only tuned the coercivity of Co layer with a thickness of 0.9 nm in Co/GdO_x system (ref. 11, 12: Nature Mater. 2015,2019). Secondly, the amplitude of coercivity modification by magneto-ionics only reaches up to a few tens of mT. Besides, the involved oxidation and change of crystal structure may make the reversibility a problem.

In our hydrogen charging/discharging concept, hydrogen atoms, created by the electrochemical reduction of water molecules, directly diffuse into the interstitial sites of bulk ferromagnetic metals and can change the magnetic properties of the whole bulk material. The driving force for the migration of hydrogen atoms is the gradient of concentration, rather than electric field as in magneto-ionics, and therefore the diffusion of hydrogen atoms is not restricted by electric field. Employing our approach, we modulated the magnetic properties of micrometer-sized SmCo₅ and changed the coercivity by ~ 1 T, two orders of magnitude larger than magneto-ionics.

To clarify the conceptual differences between magneto-ionics and our approach, the second paragraph of the introduction part is revised (page 1). Now it reads:

Recently, magneto-ionics have been employed to control the magnetic properties of ultrathin metal films. Compared with the charge-doping method, magneto-ionics use ions such as O²⁻ and H⁺, instead of electrons/holes, as the tuning agent¹⁰⁻¹⁴. For instance, in the Co (0.9 nm)/GdO_x system^{11,12}, O²⁻ ions in the ionic-conducting GdO_x layer, driven by the electric field, migrate towards/away from the GdO_x/Co interface. The change of oxidation state and crystal structure in the ultrathin Co layer permits to modify its magnetic properties. However, due to electric-field screening in the metal layer, tuning of magnetism via ionic migration is limited to the interfacial region within a few atomic layers¹⁰⁻¹³. Moreover, the modulation of coercivity reaches only a few tens of mT, thus hindering practical use of voltage-tuning effect^{11,12}. Hence, effectively tuning the magnetic properties of bulk ferromagnetic metals by small voltages is still technically challenging.

I do not wish to be overly critical here - I found this to be an interesting read and very high-quality work. It should absolutely be published in a form very similar to the present draft. I'm just not convinced it will draw broad enough interest for publication in Nature Communications as opposed to a more specialized journal.

Re: We greatly appreciate the reviewer's comments on the quality of our work. In term of broad interests, since the discovery of voltage-control of magnetism in ferromagnetic metals in 2007 (ref 6, *Science*), its application has been hampered due to two main obstacles: the effect limited to a depth of a few atomic layers at the surface and too small change of magnetic properties.

Our present work successfully overcomes both limitations. We offer a conceptually-new approach to overcome the fundamental limitations of electric-field screening in metals, enabling the voltage-control of magnetism in the bulk of metals rather than just a few monolayers at the surface. Secondly, our approach produces the giant modification of coercivity by ~ 1 T with voltages of only ~ 1 V. The change of coercivity is more than two orders of magnitudes larger than those achieved by

charge doping and magneto-ionics. The overcoming of these two obstacles has never been accessible previously. Moreover, as discussed earlier, our study is interdisciplinary and combines diverse research fields including hydrogen storage, electrochemistry, and magnetism. In our opinion an exceptionally strong magnetoelectric effect demonstrated in metal ($\Delta H_c \sim 1$ T by 1 V) resulting from the combination of conventionally diverse concepts may attract much broader readership that one can expect in a more specialized journal. With the conceptually new approach, the overcoming of the electric-field screening length, the giant tuning of coercivity, and the interdisciplinary nature of our study, we believe that it will attract high interests from those working on magnetoelectric coupling, magneto-ionics and permanent magnets.

In the event that the authors choose to revise the work, my main suggestions for the actual content of the work is that it was sometimes unclear which experiments and figures referred to which particle sizes. Thus, I was sometimes unsure how to evaluate size effects and switching speeds. I'd make it explicit as much as possible.

Re: We thank the reviewer for pointing it out. To clarify, we added one sentence to the legend of Fig. 2 in page 14: In (A) the samples had particle sizes ranging from 1 μm to 10 μm ; those in (B) were ball milled and had smaller particle sizes of ~ 1 μm .

Also, we added one sentence in page 15 to the legend of Fig. 3: Here, the same samples as in Fig. 2 A were used, with particle sizes of 1~10 μm .

One sentence was added to page 5, line 13:

In the measurement, we used the samples with particles sizes of 1~10 μm as those in Fig. 2A.

Secondly, the mass spec work shown in Figure S4 was quite important and I think it could be referred to in more detail in the main text rather than relegated to the supporting information.

Re: We appreciate the reviewer's suggestion. The mass spectroscopy showed that the deuterium concentration increased significantly after charging, confirming the absorption of hydrogen atoms in SmCo_5 . This observation directly supported and confirmed in situ XRD results (Fig. 1D, E) that showed the insertion of hydrogen atoms into the material. Therefore, we think that due to limited space available in the main text it may be reasonable to keep it in the supplementary materials to support the very convincing XRD results. To stress the importance of the mass spectroscopy results, we added one sentence to page 3, line 26 to refer to mass spectroscopy results:

Mass spectrum confirmed that after charging in 1 M KOH in D_2O , the intensity of deuterium peak increased significantly and became comparable to that of hydrogen peaks, confirming the absorption of hydrogen atoms in SmCo_5 (Fig. S4). The amount of the absorbed hydrogen was determined to be around 2.6 atoms per SmCo_5 unit cell by thermogravimetric analysis (Fig. S5).

Reviewer #2 (Remarks to the Author):

This is an interesting manuscript that reports on the effects of voltage on the magnetic properties of electrolyte-gated SmCo_5 particles (mediated through voltage-driven hydrogen absorption).

This work opens new avenues for magnetoelectric actuation of ferromagnetic materials. It can have impact in applications dealing with hard magnetic materials, spintronics, iontronics and any magnetically actuated device, where the reported drastic reduction of coercivity would significantly reduce the energy power consumption. It also explores a new mechanism to induce magnetoelectric effects in thick ferromagnetic films/particles.

Thus, the paper is of high interest for the scientific community (both from fundamental and applied viewpoints) and I recommend its publication. However, there are a few minor issues/comments the authors should address to further improve the quality of the manuscript:

Re: We greatly appreciate the highly-favorable comments of the reviewer on our work. We thank the reviewer for reading our manuscript thoroughly and offering many instructive and inspiring suggestions, most of which are reflected in the revised manuscript.

i) The authors claim that the reported procedure is suitable to tune the magnetic properties of "bulk" ferromagnets. I find this statement a bit far-fetched since they investigate the changes in the magnetic properties of micrometer-sized particles. I understand the point raised by the authors since, typically, electric field effects in metallic ferromagnets are limited to few nm thick films. Yet, for the general audience (not necessarily familiar with magnetoelectricity), to refer to micrometer-sized particles as "bulk" can be misleading. I would rather use, perhaps, "thick particles" or, at most, "bulky particles".

Re: We appreciate the reviewer's suggestion very much. As the reviewer pointed out, our approach enables the magnetism to be tuned beyond the electric-field screening length. In present manuscript, we tuned magnetic properties in the bulk of micrometer-sized SmCo_5 particles rather than only a few monolayers at the surface. However, we found it very difficult to describe concisely and briefly "not restricted to surface layers" in the title of our manuscript. In addition, the size at which materials display different properties compared with bulk materials is generally around 100 nm, and materials larger than 100 nm are often defined as bulk materials. Thus, to attract readers to notice the inherent difference of our approach to previous approaches, we think that it may be acceptable and beneficial to keep the word "bulk". To clarify what the word "bulk" refers to, in the abstract we explain its meaning by adding "beyond the electric-field screening length". Now the sentence is "open up a way to control the bulk magnetic properties of ferromagnetic metals beyond the electric-field screening length." We hope that this is an appropriate way to describe "not restricted to electric-field screening length" while eliminating confusion.

ii) Can the authors estimate how deep can hydrogen atoms be introduced (absorbed) inside the SmCo_5 particles? Could this be supported by some calculations, inferred from structural characterization, or guessed from magnetic measurements? Would the effect still work, for example, in particles of 100 microns or 1 mm size?

Re: Thanks for the questions. In the present manuscript, we used *in situ* XRD measurements in a transmission mode to characterize the evolution of crystal structure during hydrogen charging/discharging. The transmission mode allows the detection of the whole volume of the particles rather than only their surfaces. Thus, from *in situ* XRD patterns, it is clear that the whole

volume of the particles was charged with hydrogen atoms (Fig. 1D, E). (But it is only necessary to charge/discharge the near-surface region to tune the coercivity of the whole particle)

To stress the transmission mode of in situ XRD, we added one sentence in p.3, line 21 and it reads:
“The transmission mode allows the detection of the whole volume of the particles rather than only their surfaces.”

It is possible to charge SmCo₅ with larger sizes given enough time. To estimate how long it takes for charging the entire volume of particles ~100 μm, we used the diffusion equation that relates the diffusion length (l) to diffusion time (t) as described by

$$l = \sqrt{Dt} \quad (1)$$

in which D is the diffusion coefficient. As we observed, for particles with sizes of 1~10 μm, it took about 10 minutes for the full charging (Fig. 1D). Then, for a particle with a size of 100 μm, we calculate the diffusion time to be around 15 hours. However, to change the coercivity of the whole particle, it is only necessary to charge/discharge the near-surface region rather than the whole volume of the SmCo₅ magnet to tune its coercivity (bulk property), as described in paragraph 3, page 4 and by the discussion of the mechanism behind coercivity change. This would greatly reduce the tuning time. In addition, the diffusion of hydrogen atoms can be speed us by including high diffusion path such as grain boundaries.

iii) Do the authors envisage that such effects might also be observed in other hard magnetic materials? Why did they choose specifically SmCo₅, for example?

Re: Yes, the hydrogen-tuning concept can be applied to other hard magnets. Recently, we achieved a drastic modulation of coercivity in another important permanent magnet, the Sm₂Co₁₇-type magnet. We also expect the concept to be applicable to NdFeB magnet.

Based on two criteria we have evaluated the applicability of our approach and chosen the candidate magnetic materials:

1) Thermodynamic consideration. At room temperature the equilibrium pressure for hydrogen absorption and desorption in the material should be in the range of 0.1 to 5 atm. Only in this range, the equivalent electrochemical potential is suitable for electrochemically-controlled hydrogen charging/discharging. If the equilibrium pressure is too high, the electrochemical potential for hydrogen charging would be too negative, thus accompanied with massive evolution of hydrogen gas; if the equilibrium potential is too low, the charged material would be very stable and hydrogen atoms cannot be discharged. Regarding SmCo₅ at room temperature its equilibrium hydrogen pressure is 4 atm, thus suitable for electrochemically-controlled charging/discharging with hydrogen atoms.

2) Materials with useful - in the context of applications - magnetic properties. As demonstrated in our manuscript, SmCo₅ has the hitherto highest magnetocrystalline anisotropy and is promising for use in next-generation ultrahigh density magnetic recording. But the high magnetocrystalline anisotropy leads to high coercivity, making the reversal (or the writing) of magnetic bits difficult. Thus, the tuning of its coercivity and the voltage-assisted magnetization reversal, as demonstrated in this work,

would be promising for future applications. In addition, SmCo₅ is an important permanent magnet. During the handling and processing of the magnet, a temporary low-coercivity state is suitable for demagnetization and further magnetizing, which significantly reduce the energy consumption.

To reflect the discussion here, we have revised part of the third paragraph in the introduction to:

To test the idea, we selected SmCo₅ as a model material based on two criteria. Firstly, its equilibrium hydrogen pressure is 4 atm at room temperature¹⁸ and according to Nernst equation, it is calculated that the equivalent electrochemical potential is only ~17 mV more negative than the standard water electrolysis potential, making SmCo₅ suitable for voltage-controlled hydrogen charging/discharging. Secondly, SmCo₅ is widely used as an important permanent magnet for its large coercivity, large magnetization (100 A m²/kg) and high Curie temperature (1020 K)¹⁹ and is considered candidate material in next-generation ultrahigh density magnetic storage (area density ~ 10 TB/inch²) because of its exceptionally high magnetocrystalline anisotropy (~17.4 MJ/m³)²⁰. Tuning of its magnetic properties, particularly magnetocrystalline anisotropy and coercivity with small voltages, would create novel magneto-electric functions in application context.

iv) Such hard magnetic materials contain rare-earths (Sm in this case), which are rather prone to oxidation. Did the samples become partially oxidized during the magnetoelectric experiments? Did the authors use some procedure to prevent from oxidation? Why there is apparently no oxidation? The authors could explain this in terms of the voltage values applied.

Re: We thank the reviewer for the question. As the reviewer pointed out, the applied electrochemical potential is in the range of reduction for SmCo₅ that prevents its oxidation in electrolyte. Moreover, the element samarium, after mixing with Co, is much less prone to oxidation. The bulk SmCo₅ remains shining and its magnetization remains constant at ambient conditions for months.

v) Can the authors comment a bit more on the dynamics of the system (for example, for the cycling experiments)? Being H atoms very light, I might be expecting faster rates (as in the works by Prof. Beach -although those were for ultra-thin films). The magneto-ionic rates here seem comparable to oxygen or lithium ion intercalation. Can the authors compare with other typical magneto-ionic rates on thick films?

Re: As far as we know, the fastest switching speed of magneto-ionics is achieved by Prof. Geoffrey Beach's group, and, therefore, we refer to their work to make comparison with our approach. In their papers (ref. 11,12), for Co/GdO_x system with Co layer of 0.9 nm, to tune the coercivity by 10~ 20 mT, the tuning time with O²⁻ is 100~ 200 s at 100 °C (Fig. 2, 3 in ref. 11). By increasing the temperature or engineering the electrode and GdO_x, the tuning time can be reduced to 10 ms at 120 °C (Fig. 3f in ref. 11) or to 700 μs at 100 °C (Fig. 4i in ref. 11). By using the voltage-driven migration of H⁺, for the GdO_x/Co system with Co thickness of 0.9 nm, the so-far achieved fastest tuning time is 100-400 ms at room temperature (Supplementary Fig. 10 in ref. 12). The switching speed of magneto-ionics is mainly limited by the slow diffusion of ions in the conducting oxide.

Using the same diffusion equation ($l = \sqrt{Dt}$) as in Response to Question ii), we estimated the

diffusion time of hydrogen atoms in SmCo₅ for a thickness of 1 nm. In our case of SmCo₅ particle with sizes of 1~10 μm, the full charging time is ~ 10 minutes (Fig. 1D). For estimation, we use the charging time of 10 minutes and the particle size of 5 μm. Then, assuming the same diffusion coefficient, it is calculated that for a thickness of 1 nm, the diffusion time is approximately 2.4×10^{-5} seconds (0.024 ms) according to equation [1]. Therefore, with the comparable thickness (~1 nm), the tuning speed of our approach should be at least three orders of magnitudes faster than magneto-ionics at room temperature (100-400 ms in ref 12). Even if we use the much longer discharging time of 40 hours to calculate, the diffusion time is 5.8×10^{-3} seconds, still at least one order of magnitude faster than magneto-ionics at room temperature.

In the present manuscript, we mainly demonstrate that using our approach, we can tune magnetism in the bulk of ferromagnetic metals and more detailed study of tuning speed and hydrogen dynamics will certainly be carried out in future studies. To reflect the comparison of the tuning speed of our approach with that of magneto-ionics, we have added a few sentences to the conclusion of the manuscript (page 7, line 28) and it reads:

It is anticipated that the hydrogen charging/discharging time can be significantly reduced when the material size is reduced to nanometer scale and the switching speed can greatly increase. To estimate, we calculated the diffusion time in SmCo₅ with a thickness of 1 nm according to the diffusion equation $l = \sqrt{Dt}$, in which l is the diffusion length, D diffusion coefficient and t diffusion time. In Fig. 1D, the charging time of SmCo₅ particles with sizes of 1~10 μm is about 10 minutes, and then the diffusion time for a thickness of 1 nm is calculated to be in the order of 0.01 ms at room temperature, at least three orders of magnitudes faster than magneto-ionics.

vi) Can the initial pristine coercivity be recovered? Why not? It seems there is some irreversible structural change in the particles the first time voltage is applied. Can the authors further comment on that?

Re: Yes, the initial pristine coercivity can be completely recovered as shown in Fig. 2A and Fig. S7. To make it clear, we revised page 4, line 23 to “When the sample was fully discharged (S_{FDC}), both the magnetization and the coercivity of the sample were fully recovered to their initial values of the pristine (S_p) sample (Fig. 2A, fig. S7).” Further, we added one sentence to the legend of Fig. 2A and it reads: The fully-discharged sample (S_{FDC}) was obtained by discharging the S_{FC} sample at -0.4 V for ~ 40 hours, showing the complete recovery of the coercivity to that of S_p sample.

vii) One of the intriguing aspects of the work (also pointed out by the authors) is why the coercivity changes occur much faster than those of the magnetization when the sample is subject to applied voltage. Besides the explanations provided by the authors (which seem plausible to me, in spite of the complexity of the effects), could it be that there is some exchange-spring mechanism going on here as well, similar to that of "spring magnets" but inducing softening (rather than hardening)?

Re: We thank the reviewer for pointing out this possibility. As the reviewer said, after partial discharging, the near-surface region of the SmCo₅ particle has larger magnetocrystalline anisotropy

while the interior of the particle has lower values. At the interface, there may be a coupling between the magnetically-hard shell and the magnetically-soft core, forming the exchange spring at the interface. Yet, we think that the exchange-spring may not play a critical role in determining the coercivity. Firstly, the hydrogen concentration changes gradually from the surface of the particle to the core and there is no sharp interface between the core and the shell. Secondly, in exchange-spring systems, the length scale of the soft/hard components should be comparable to the exchange correlation length (or domain wall width), $\delta = \pi\sqrt{A/K}$ in which A and K are the exchange and anisotropy constants, respectively. For SmCo_5 , the exchange correlation length is 2~4 nm (see references 1-3 listed below). The exchange-spring only dominates in nanocomposite and multilayers, in which the particle size or the layer thickness is a few nanometers. However, in our case the particle size is in the micrometer range, we therefore consider the exchange-spring effect not to play a significant role.

Ref. 1: E. Fullerton, J.S. Jiang, S.D. Bacher. Hard/soft magnetic heterostructures: model exchange-spring magnets. *J. Magn. Magn. Mater.* 200, 392-404 (1999).

Ref. 2: E. F. Kneller, R. Hawig. The exchange-spring magnet: a new material principle for permanent magnets. *IEEE Transac. Magn.* 27, 3588-3600 (1991).

Ref. 3: J.M.D. Coey, *Magnetism and magnetic materials*. (Cambridge University Press, Cambridge, 2009), p.266.

To reflect the discussion here, we added one sentence to page 6, line 38:

Exchange spring effect may also be involved at the soft core/hard shell interfaces. Yet, considering the gradual change of hydrogen concentration from the shell to the core as well as the rather small exchange length of SmCo_5 (2~4 nm), the exchange spring effect may be insignificant.

Accordingly, one reference (ref 32) was added:

32. E. Fullerton, J.S. Jiang, S.D. Bacher. Hard/soft magnetic heterostructures: model exchange-spring magnets. *J. Magn. Magn. Mater.* **200**, 392-404 (1999).

Reviewers' Comments:

Reviewer #1:

Remarks to the Author:

In my last report, I raised concerns regarding the suitability of the manuscript for Nature Communications based on my impression that the work does not represent a sufficiently significant advance in the field. That being said, it's certainly something that I personally found very interesting. I also noted the overall high quality and convincing nature of the experimental work.

In response, the authors made two arguments for impact:

A. The authors summarize the work stating that "by electrochemically-controlled hydrogen charging/discharging, we have successfully overcome the electric-field screening in metals and produced giant magnetoelectric effect"

B. The authors estimated the response times associated with scaling down the technique to ultrathin films, arriving at response times they state are expected to be "at least three orders of magnitude that magneto-ionics".

Unfortunately, while the authors make many good points in their very thorough response, the additions to the text have resulted in the introduction of a number of issues:

1. Both reviewers objected to the use of the term "bulk" to describe the particles used in this experiment, and I continue to feel that this will be confusing to readers. There is a difference between materials properties becoming "bulk-like" at 100 nm and actual "bulk" materials. I don't see any reason, other than appearing more impactful, that a more accurate and descriptive term like "micron-scale" couldn't be employed throughout the work.
2. The comparison to the state-of-the-art magneto-ionic switching times is not correct. For comparison, the manuscript references Nature Materials 14 (2), 174 and Nature materials 18 (1), 35. The current world record magneto-ionic switching speed (as far as I am aware) was actually demonstrated this April in Nano Letters 20, 3435 (2020) to be ~ 1 ms for both charging and discharging. This is approximately 50 times slower than the authors calculated charging speed and 5 times faster than the calculated discharging speed. Clearly these additions to the manuscript must be revised, and the statements about theoretical benefits greatly softened.
3. The calculation methods used to estimate charge/discharge times in nanoscale thin films are far too simplistic to be useful. Although I will admit a lack of expertise on electrolyte response time, I think that any such estimates (if included in the manuscript, which they don't necessarily need to be) ought to be much more detailed and include an analysis of electrolyte response time.

More minor issues:

4. The comparisons regarding coercivity tuning are likely to confuse readers - yes, the magnitude of the coercivity change demonstrated by the authors is much larger than traditional Co-based magneto-ionics, but it is the change in easy axis direction that provides the switching effect. So this is not really an accurate figure of merit.
5. The ability to reversibly tune materials beyond the screening length is indeed significant. I will note that Gilbert et al. (Nature Communications 7, 12264) have demonstrated the ability to magneto-ionically tune thicker Co films with some degree of reversibility, although the lack of complete reversibility in that study tends to support the author's point that metallic screening is a significant barrier to implementation. Still, I think that the ability of magneto-ionics to address this issue may not be as nonexistent as the authors imply.

All of these things being said, the author's points about being able to switch the coercivity at the

micron-scale (and potentially beyond) through a hydrogen insertion very near the surface is very important and DOES point towards future work in which true bulk systems might be affected. I also generally found the arguments about overcoming screening effects convincing, even if some of the specifics were objectionable. On balance, the authors have convinced me to change my position on the impact and suitability of this manuscript for Nature Communications, and I would support publication if the authors are willing to address issues 1-5 (or at the very least 1-3) above.

Reviewer #2:

Remarks to the Author:

The authors have satisfactorily addressed my previous criticism. Given the quality of the work and the clarity of the manuscript, I believe it can now be accepted for publication in Nature Communications in its present form.

REVIEWER COMMENTS

Reviewer #1 (Remarks to the Author):

In my last report, I raised concerns regarding the suitability of the manuscript for Nature Communications based on my impression that the work does not represent a sufficiently significant advance in the field. That being said, it's certainly something that I personally found very interesting. I also noted the overall high quality and convincing nature of the experimental work.

In response, the authors made two arguments for impact:

A. The authors summarize the work stating that "by electrochemically-controlled hydrogen charging/discharging, we have successfully overcome the electric-field screening in metals and produced giant magnetoelectric effect"

B. The authors estimated the response times associated with scaling down the technique to ultrathin films, arriving at response times they state are expected to be "at least three orders of magnitude that magneto-ionics".

Unfortunately, while the authors make many good points in their very thorough response, the additions to the text have resulted in the introduction of a number of issues:

1. Both reviewers objected to the use of the term "bulk" to describe the particles used in this experiment, and I continue to feel that this will be confusing to readers. There is a difference between materials properties becoming "bulk-like" at 100 nm and actual "bulk" materials. I don't see any reason, other than appearing more impactful, that a more accurate and descriptive term like "micron-scale" couldn't be employed throughout the work.

2. The comparison to the state-of-the-art magneto-ionic switching times is not correct. For comparison, the manuscript references Nature Materials 14 (2), 174 and Nature materials 18 (1), 35. The current world record magneto-ionic switching speed (as far as I am aware) was actually demonstrated this April in Nano Letters 20, 3435 (2020) to be ~ 1 ms for both charging and discharging. This is approximately 50 times slower than the authors calculated charging speed and 5 times faster than the calculated discharging speed. Clearly these additions to the manuscript must be revised, and the statements about theoretical benefits greatly softened.

3. The calculation methods used to estimate charge/discharge times in nanoscale thin films are far too simplistic to be useful. Although I will admit a lack of expertise on electrolyte response time, I think that any such estimates (if included in the manuscript, which they don't necessarily need to be) ought to be much more detailed and include an analysis of electrolyte response time.

More minor issues:

4. The comparisons regarding coercivity tuning are likely to confuse readers - yes, the magnitude of the coercivity change demonstrated by the authors is much larger than traditional Co-based magneto-ionics, but it is the change in easy axis direction that provides the switching effect. So this is not really an accurate figure of merit.

5. The ability to reversibly tune materials beyond the screening length is indeed significant. I will note that Gilbert et al. (Nature Communications 7, 12264) have demonstrated the ability to magneto-ionically tune thicker Co films with some degree of reversibility, although the lack of complete reversibility in that study tends to support the author's point that metallic screening is a significant barrier to implementation. Still, I think that the ability of magneto-ionics to address this issue may not be as nonexistent as the authors imply.

All of these things being said, the author's points about being able to switch the coercivity at the micron-scale (and potentially beyond) through a hydrogen insertion very near the surface is very important and DOES point towards future work in which true bulk systems might be affected. I also generally found the arguments about overcoming screening effects convincing, even if some of the specifics were objectionable. On balance, the authors have convinced me to change my position on the impact and suitability of this manuscript for Nature Communications, and I would support publication if the authors are willing to address issues 1-5 (or at the very least 1-3) above.

Reviewer #2 (Remarks to the Author):

The authors have satisfactorily addressed my previous criticism. Given the quality of the work and the clarity of the manuscript, I believe it can now be accepted for publication in Nature Communications in its present form.

Reply to reviewers' comments

Reviewer #1 (Remarks to the Author):

In my last report, I raised concerns regarding the suitability of the manuscript for Nature Communications based on my impression that the work does not represent a sufficiently significant advance in the field. That being said, it's certainly something that I personally found very interesting. I also noted the overall high quality and convincing nature of the experimental work.

Re: We thank the reviewer again for the favorable comments on the quality of our research. We are glad that we are able to convince her/him "to change my position on the impact and suitability of this manuscript for Nature Communications, and I would support publication if the authors are willing to address issues 1-5 (or at the very least 1-3) above." In the following we addressed all 1-5 issues, and revised the manuscript accordingly.

In response, the authors made two arguments for impact:

A. The authors summarize the work stating that "by electrochemically-controlled hydrogen charging/discharging, we have successfully overcome the electric-field screening in metals and produced giant magnetoelectric effect"

B. The authors estimated the response times associated with scaling down the technique to ultrathin films, arriving at response times they state are expected to be "at least three orders of magnitude that magneto-ionics".

Unfortunately, while the authors make many good points in their very thorough response, the additions to the text have resulted in the introduction of a number of issues:

Re: We understand that the reviewer has concerns about using the word "bulk" in the manuscript as well as the estimated switching time at nanoscale thin films. Below we address them in detail.

1. Both reviewers objected to the use of the term "bulk" to describe the particles used in this experiment, and I continue to feel that this will be confusing to readers. There is a difference between materials properties becoming "bulk-like" at 100 nm and actual "bulk" materials. I don't see any reason, other than appearing more impactful, that a more accurate and descriptive term like "micron-scale" couldn't be employed throughout the work.

Re: Indeed we introduced the term "bulk" to emphasize that our approach overcomes the electric-field screening and allows the tuning of magnetic properties deep in the volume of the material. However, we accept the reviewer's argument that the

“micron-scale” is more precise here and simply describes the actual experimental conditions i.e. the size of the hydrogen-charged SmCo₅ particles. We accepted the reviewer’s suggestion and replaced the word “bulk” by “micron-scale” in the manuscript.

Changes made to the manuscript:

page 1, Title: Giant Voltage-Induced Modification of Magnetism in **Micron-Scale** Ferromagnetic Metals by Hydrogen Charging

page 1, Abstract: **Our study opens up a way to control the magnetic properties in ferromagnetic metals beyond the electric-field screening length,**

Similarly, when appropriate, other references to “bulk” were either removed or replaced by “micron-scale” in the revised manuscript.

2. The comparison to the state-of-the-art magneto-ionic switching times is not correct. For comparison, the manuscript references Nature Materials 14 (2), 174 and Nature materials 18 (1), 35. The current world record magneto-ionic switching speed (as far as I am aware) was actually demonstrated this April in Nano Letters 20, 3435 (2020) to be ~ 1 ms for both charging and discharging. This is approximately 50 times slower than the authors calculated charging speed and 5 times faster than the calculated discharging speed. Clearly these additions to the manuscript must be revised, and the statements about theoretical benefits greatly softened.

Re: We thank the reviewer for pointing out the reference. In this reference, by replacing GdO_x with high proton-conductive yttria-stabilized zirconia Lee et al. decreased the switching time of magneto-ionics to ~ 1 ms at room temperature in ultrathin films of cobalt (1 nm). Being aware of this reference, we revised the text to (page 7, line 31):

“We thus estimated the switching speed at the nanoscale by calculating the diffusion time according to the diffusion equation, $l = \sqrt{Dt}$, in which l is the diffusion length, D the diffusion coefficient and t the diffusion time. In Fig. 1D, the charging/discharging time of SmCo₅ particles with sizes of 1~10 μm is ~10 minutes/40 hours. Therefore, the diffusion time for a thin film at the nanometer scale can be expected to be reduced by several orders of magnitudes to ms and sub-ms range, **which is comparable to the fastest switching speed (~1 ms for 1 nm-thick cobalt layer³⁸) achieved by magneto-ionics at similar length scales.”**

Reference added:

41. K.Y. Lee, S. Jo, A. Tan, M. Huang, D. Choi, J. H. Park, H. Ji, J. W. Son, J. Chang, G. Beach, S. Woo. Fast Magneto-Ionic Switching of Interface Anisotropy Using Yttria-Stabilized Zirconia Gate Oxide. *Nano Lett.* **20**, 3435-3441 (2020).

3. The calculation methods used to estimate charge/discharge times in nanoscale thin films are far too simplistic to be useful. Although I will admit a lack of expertise on electrolyte response time, I think that any such estimates (if included in the manuscript, which they don't necessarily need to be) ought to be much more detailed and include an analysis of electrolyte response time.

Re: As the reviewer has correctly noted, an attempt to estimate hydrogen diffusion time at nanoscale by rescaling values from the microscale to nanoscale is burdened with uncertainties difficult to appraise. However, definitely not putting undue emphasis on this aspect of our work, we think that it would be of benefit to the reader to get an assessment of the hydrogen-charging kinetics. This would help the research community to put the present work in the broader context and get some rough feeling for the possible research directions and application prospects.

To support our point of view, we hope that we can convince the reviewer that an electrolyte (ionic liquid), in terms of its conductivity (or ion transportation speed), is not a limiting factor in the voltage-driven hydrogen charging. In the well documented research (including work done in our group) on electrolyte-gated printable electronics, it has been shown experimentally that using electrolyte and ionic liquids in the optimized geometries makes it possible to reach switching times in the kHz (ms) range. The theoretical switching frequencies of electrolyte-gated electronics are predicted in MHz range.

Therefore, the hydrogen adsorption at the surface and its consequent diffusion into the metal interior metal may be a limiting factor. As the reviewer pointed out, the hydrogen charging process is composed of two steps (Fig. 1B): the 1st step with the production of hydrogen atoms from electrochemical reduction of water molecules at metal surface and the 2nd step with the subsequent diffusion of hydrogen atoms into the metal structure. Usually, with the application of large negative over-potential, the hydrogen atoms can be produced easily at the metal surface, and therefore, the hydrogen diffusion in the material is the rate-limiting step of the switching speed. Thus, we used the hydrogen diffusion time to estimate the switching speed. Yet, we agree with the reviewer that at the nanometer scale the kinetics of these two steps may be varied and the calculation may become too simplistic. On the other hand, the effective diffusion coefficient that we estimate has its own merit by the virtue of being obtained experimentally. According to the diffusion equation, $l = \sqrt{Dt}$, using ~10 minutes/40 hours for the respective charging/discharging time in 1~10 μm SmCo5 particles (Fig. 1D), the diffusion coefficient is calculated in the range of 10^{-8} ~ 10^{-13} cm^2/s at room temperature. These values are still much smaller than that obtained in gaseous hydrogenation (10^{-8} ~ 10^{-10} cm^2/s), indicating that significant improvements may be achieved by optimizing the electrochemical-device geometry. Considering all the above points, we have revised the estimation and now describe the estimation in a more qualitative and conservative way. It reads (page 7, line 29):

“It is anticipated that the hydrogen charging/discharging time can be significantly reduced when the material size is reduced to nanometer scale and the switching speed can be increased. The production of hydrogen atoms by electrochemical reduction of water molecules is considered much faster than the diffusion of hydrogen atoms in the material. We thus estimated the switching speed at the nanoscale by calculating the diffusion time according to the diffusion equation, $l = \sqrt{Dt}$, in which l is the diffusion length, D the diffusion coefficient and t the diffusion time. In Fig. 1D, the charging/discharging time of SmCo₅ particles with sizes of 1~10 μm is ~10 minutes/40 hours. Therefore, the diffusion time for a thin film at the nanometer scale can be expected to be reduced by several orders of magnitudes to ms and sub-ms range, which is comparable to the fastest switching speed (~1 ms for 1 nm-thick cobalt layer³⁸) achieved by magneto-ionics at similar length scales. In addition, based on this equation, the calculated diffusion coefficient falls in the range of 10⁻⁸~10⁻¹³ cm²/s at room temperature, still much smaller than that obtained in gaseous hydrogen (10⁻⁸~10⁻¹⁰ cm²/s^{42,43}), indicating that significant improvements in switching speed may be achieved by optimizing the electrochemical-cell (device) geometry^{44,45}.”

References added:

42. D. Richter, R. Hempelmann, L.A. Vinhas. Hydrogen Diffusion in LaNi₅H₆ Studied by Quasi-Elastic Neutron Scattering. *J. Less-Common Met.* **88**, 353-360 (1982).
43. L. Ming, E. Lavendar, A.J. Goudy. The Hydriding and Dehydriding Kinetics of Some RCo₅ Alloys. *Int. J. Hydrogen Energy* **22**, 63-66 (1997).
44. C. Mackin, E. McVay, T. Palacios. Frequency Response of Graphene Electrolyte-Gated Field-Effect Transistors. *Sensors* **18**, 494 (2018).
45. M. J. Ha, Y. Xia, A.A. Green, W. Zhang, M.J. Renn, C.H. Kim, M. Hersam, C. D. Frisbie. Printed, Sub-3 V Digital Circuits on Plastic from Aqueous Carbon Nanotube Inks. *ACS Nano* **4**, 4388-4395 (2010).

More minor issues:

4. The comparisons regarding coercivity tuning are likely to confuse readers - yes, the magnitude of the coercivity change demonstrated by the authors is much larger than traditional Co-based magneto-ionics, but it is the change in easy axis direction that provides the switching effect. So this is not really an accurate figure of merit.

Re: We agree with the reviewer that the mechanism behind the coercivity change is completely different between magneto-ionics and our approach. The tuning mechanism of charge-doping method is also different. However, despite the different mechanism behind the switching effect, the goal of all these works is to achieve large switching effects. Hence, from a perspective of applications, we believe it may be

acceptable and justifiable to make such a comparison of the strength of the magnetic response to the electric stimulus, regardless of the underlying mechanism.

5. The ability to reversibly tune materials beyond the screening length is indeed significant. I will note that Gilbert et al. (Nature Communications 7, 12264) have demonstrated the ability to magneto-ionically tune thicker Co films with some degree of reversibility, although the lack of complete reversibility in that study tends to support the author's point that metallic screening is a significant barrier to implementation. Still, I think that the ability of magneto-ionics to address this issue may not be as nonexistent as the authors imply.

Re: We thank the reviewer for the comment. To reflect the efforts to tune magnetism on a large scale by magneto-ionics, (the referred Gilbert et al. (Nature Communications 7, 12264 would be an example of an electrochemical redox, conversion-type reaction), we have added to the introduction:

Although tuning of metallic layer with larger thickness (~15 nm) has also been achieved by magneto-ionics, these tuning processes often suffer from the inherent irreversibility, typical of electrochemical conversion-type reactions¹⁵.

Reference added:

15. D. A. Gilbert, A. J. Grutter, E. Arenholz, K. Liu, B. J. Kirby, J. A. Borchers, B. B. Maranville. Structural and magnetic depth profiles of magneto-ionic heterostructures beyond the interface limit. *Nat. Commun.* **7**, 12264 (2016).

All of these things being said, the author's points about being able to switch the coercivity at the micron-scale (and potentially beyond) through a hydrogen insertion very near the surface is very important and DOES point towards future work in which true bulk systems might be affected. I also generally found the arguments about overcoming screening effects convincing, even if some of the specifics were objectionable. On balance, the authors have convinced me to change my position on the impact and suitability of this manuscript for Nature Communications, and I would support publication if the authors are willing to address issues 1-5 (or at the very least 1-3) above.

Re: We are very grateful that the reviewer is convinced about the potential impact of our manuscript and its suitability to publish in Nature Communications. We hope that the reviewer will find our adjustments to the manuscript related to the issues 1-5 appropriate and convincing.

Reviewer #2 (Remarks to the Author):

The authors have satisfactorily addressed my previous criticism. Given the quality of the work and the clarity of the manuscript, I believe it can now be accepted for

publication in Nature Communications in its present form.

Re: We are very grateful that the reviewer offered highly-favorable comments on our work and providing insightful and useful suggestions that greatly helped strengthen our manuscript.

Reviewers' Comments:

Reviewer #1:

Remarks to the Author:

I would like to thank the authors for their comprehensive and detailed responses to my comments in both rounds of review. All of my concerns have been addressed, and I support publication of the present form of the manuscript in Nature Communications.

REVIEWERS' COMMENTS:

Reviewer #1 (Remarks to the Author):

I would like to thank the authors for their comprehensive and detailed responses to my comments in both rounds of review. All of my concerns have been addressed, and I support publication of the present form of the manuscript in Nature Communications.

Re: We greatly appreciate the reviewer's favorable comments on the quality of our work. We also thank the reviewer very much for proving insightful and critical suggestions that helped convey the central message of our manuscript.